# What, when, and where? - Self-Supervised Spatio-Temporal Grounding in Untrimmed Multi-Action Videos from Narrated Instructions

## Abstract

Spatio-temporal grounding describes the task of localizing events in space and time, e.g., in video data, based on verbal descriptions only. Models for this task are usually trained with human-annotated sentences and bounding box supervision. This work addresses this task from a multimodal supervision perspective, proposing a framework for spatio-temporal action grounding trained on loose video and subtitle supervision only, without human annotation. To this end, we combine local representation learning, which focuses on leveraging fine-grained spatial information, with a global representation encoding that captures higher-level representations and incorporates both in a joint approach. To evaluate this challenging task in a real-life setting, a new benchmark dataset is proposed providing dense spatio-temporal grounding annotations in long, untrimmed, multi-action instructional videos for over $5K$ events. We evaluate the proposed approach and other methods on the proposed and standard downstream tasks showing that our method improves over current baselines in various settings, including spatial, temporal, and untrimmed multi-action spatio-temporal grounding.

## 1 Introduction

Spatio-temporal grounding (STG) describes the challenging task of locating events in space and time within video data based on text referential expressions. Methods in this field usually rely on a combination of spatio-temporal bounding box annotation, together with a human-generated caption, describing the visual content of the bounding box (Yang et al., 2022a; Jin et al., 2022), which limits their generalizability beyond the given training scenario. Compared to that, multimodal self-supervised learning tries to leverage "free" data sources, such as video and automatic speech recognition (ASR) captions from large-scale instructional videos to learn representations without human annotation (Miech et al., 2019; 2020; Alayrac et al., 2020; Akbari et al., 2021; Chen et al., 2021). The resulting models achieve state-of-the-art performance on zero-shot tasks such as cross-modal video retrieval or classification and also for zero-shot temporal action segmentation and detection based on free text queries (Zhukov et al., 2019; Kuehne et al., 2019; Tang et al., 2019; Chen et al., 2021; Shvetsova et al., 2022), but usually lack spatial localization abilities. Other approaches focus on label-free spatial grounding, e.g. by training on image-caption (Akbari et al., 2019; Yang et al., 2022b; Li et al., 2022a; Wang et al., 2022; Zhong et al., 2022) or video-caption pairs (Tan et al., 2021; Shi et al., 2019). The goal is to correctly localize a referential expression in an image or each video frame, e.g., via a bounding box or a heatmap. However, those methods are not optimized to detect whether an event is present in a video. The assumption is thus that the evaluated expression is visible in the image or in all video frames.

The following work aims to bring together those ideas to address the task of spatio-temporal action grounding from multimodal supervision in untrimmed videos. We propose a grounding approach that uses video-text pairs based on ASR transcripts in instructional videos and learns the spatial representation of free-text events as well as their temporal extent, as shown in Figure 1. To this end, we leverage two different representations of the visual data: a global feature representation based on full-frame information to define the temporal extent of an event and a local representation based

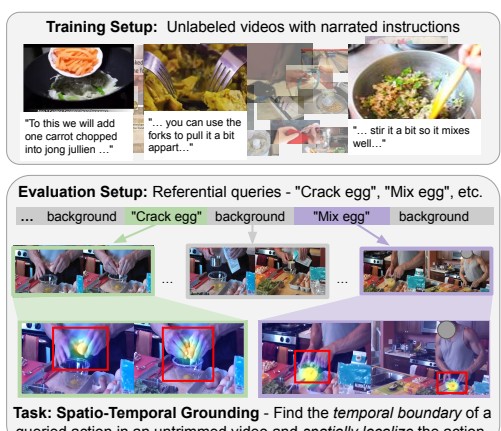

Figure 1: **Learning Spatio-temporal grounding in untrimmed videos.** In training, we learn from unlabeled videos without human annotation. In evaluation, we perform spatio-temporal grounding using an action description such as "crack egg" as a query. The model needs to localize both the action's temporal boundary and spatial region in the long untrimmed video. We visualize the heat-map from the annotation points as well as derived bounding boxes.

| Dataset | Annotation | |
|---|---|---|
| | Spatial | Temporal |
| V-HiCo | object bb + human bb | - |
| AVA-Kinetics | object bb + human bb | - |
| THUMOS14 | - | action boundaries |
| ActivityNet | - | action boundaries |
| HACSSegment | - | action boundaries |
| YouCook II | - | multi-action boundaries |
| Cross-Task | - | multi-action boundaries |
| COIN | - | multi-action boundaries |
| EPIC KITCHENS-100 | - | multi-action boundaries |
| Ego4D | - | multi-action boundaries |
| JHMDB51-21 | human tubes | - |
| UCF101-24 | human tubes | action boundaries |
| Daly | human tubes | action boundaries |
| Vid-STG | human tubes | action boundaries |
| HC-STVG | human tubes | action boundaries |
| AVA | human tubes | action boundaries |
| YouCook-Interactions | action bb | - |
| GroundingYoutube (ours) | action bb + center points | multi-action boundaries |

Table 1: **Comparison of spatial, temporal, and spatio-temporal grounding datasets.** V-HiCo (Li et al., 2021), AVA-Kinetics (Li et al., 2020), THUMOS14 (Idrees et al., 2017), ActivityNet (Caba Heilbron et al., 2015), HACSSegment (Zhao et al., 2019), YouCook II (Zhou et al., 2018b), Cross-Task (Zhukov et al., 2019), COIN (Tang et al., 2019), EPIC KITCHENS-100 (Damen et al., 2022), Ego4D (Grauman et al., 2022), JHMDB51-21 (Jhuang et al., 2013), UCF101-24 (Soomro et al., 2012) Daly (Weinzaepfel et al., 2016), Vid-STG (Zhang et al., 2020), HC-STVG (Tang et al., 2021), AVA (Gu et al., 2018), YouCook-Interactions (Tan et al., 2021).

on frame-wise grid features for spatial localization. The motivation for this dualism is that while the local representation captures the spatial correlations between vision and text input, this can be too fine-grained to learn a holistic representation of the frame, while the global representation can be assumed to capture a more compact, aggregated view compared to local data and thus to provide a more reliable cue for the task of temporal localization. However, compared to the hand-annotated video-caption setup of most spatio-temporal grounding methods, the ASR text can be noisy as not all comments refer to visible events. Further, as there is only a loose temporal correlation, the described activities might not be precisely aligned, can be scattered over multiple frames, or not be present at all (Miech et al., 2020; Han et al., 2022). Therefore, we propose to specifically select frames to capture only those useful for training. To this end, we look for frames that match the vocabulary of the respective text, leveraging a selection strategy by Sinkhorn optimal transport (Cuturi, 2013). This allows us to train a model that can localize actions in space and time within videos without labeling supervision.

To evaluate spatio-temporal grounding in untrimmed videos, a new benchmark, GroundingYouTube, is proposed. It is based on the existing MiningYouTube dataset (Kuehne et al., 2019) and extended with spatio-temporal localization information. This setup differs from other benchmarks such as (Tan et al., 2021; Chen et al., 2019; Zhou et al., 2018a) in two ways: first, by using multiple center point annotations, it focuses on the grounding of referential actions itself instead of interacting humans or objects which are usually labeled; second, the dense annotations of multiple actions in the video allow us to benchmark action grounding in long, realistic untrimmed videos compared to existing, often pre-clipped benchmarks(Chen et al., 2019; Zhang et al., 2020). The benchmark provides queries for 512 different event types and over $5K$ spatio-temporal annotations as shown in Figure 1. A comparison of current datasets is shown in Table 1.

To evaluate the proposed approach as well as the new benchmark, the system is trained on the HowTo100M dataset (Miech et al., 2019) and compared to state-of-the-art methods based on full, weak, and self-supervision for spatial and temporal, as well as combined spatio-temporal grounding tasks. It shows that existing methods usually do well in one of the two aspects, spatial or temporal grounding. In contrast, the proposed method can combine spatial and temporal aspects of semantic concepts without label annotation.

We summarize the contributions of this work as follows[1]: (1) We propose a framework for spatio-temporal grounding in untrimmed videos based on weakly aligned multimodal supervision without human annotation, employing a combination of global and local representation learning to learn the spatio-temporal extent of actions. (2) To facilitate this task, we propose a frame selection strategy based on Sinkhorn-Knopp Optimal transport that improves the quality of the acquired learning samples, leading to more effective supervision. (3) We provide a new benchmark and annotations to evaluate this challenging problem on real-world multi-action instructional video data.

## 2 RELATED WORK

**Supervised Spatio-temporal Grounding.** Spatio-temporal Grounding refers to the problem of localizing a sequence of bounding boxes (a spatio-temporal tube) for a target object described by an input text. This problem has been addressed by various approaches TubeDETR (Yang et al., 2022a), STCAT (Jin et al., 2022), STVGBert (Su et al., 2021), STVGBert (Su et al., 2021), STGVT (Tang et al., 2021), STGRN (Zhang et al., 2020). These methods rely on proposal networks such as Faster R-CNN (Ren et al., 2015) or MDETR (Kamath et al., 2021) to predict bounding box coordinates for learning text-to-region interaction. All those approaches rely on supervised training with the human-annotated sentence and bounding box supervision, provided, e.g., by datasets such as Vid-STG (Zhang et al., 2020) and HC-STVG (Chen et al., 2019). While those datasets provide a temporal aspect, temporal detection is usually limited to identifying the start and end frame of a single action in a video. Compared to that, an untrimmed setting usually comprises multiple actions in a longer video that can be separated by longer background sequences. This conceptually differs from previous works (Chen et al., 2019) that typically use short videos of around 5-10 seconds.

**Multimodal Self-supervised Learning.** The field of multimodal self-supervised learning aims to learn data representations by leveraging large amounts of unlabeled data with multiple modalities. Early works (Weston et al., 2011; Frome et al., 2013) started by projecting images and text into a joint visual-language embedding space, where embeddings of semantically similar pairs are close. Those ideas have now grown into systems such as MIL-NCE (Miech et al., 2020) using the HowTo100M dataset (Miech et al., 2019) to train a video-language embedding space from 1.2 million instructional videos paired with text descriptions from ASR. Follow-up works, including (Alayrac et al., 2020; Akbari et al., 2021; Rouditchenko et al., 2020; Chen et al., 2021; Shvetsova et al., 2022) show that using videos without annotation enables an effective multimodal embedding space via contrastive learning.

Based on those advantages, approaches started to address the problem of **Spatial Video Grounding** from multimodal self-supervised aiming to identify spatial locations in a *trimmed* video based on text descriptions without the need for bounding box annotation during training. One of the early works studied this task in the context of weakly supervised learning scenarios where we learn grounding with human-annotated captions of the video (Zhou et al., 2018a). In this context, works (Tan et al., 2021; Shi et al., 2019) have focused on object grounding benchmarks such as YouCook2-BoundingBox (Zhou et al., 2018b), which provides bounding box annotations for visible objects in cooking videos. Other works such as GLIP (Li et al., 2022a) and RegionCLIP (Zhong et al., 2022) combine the principles of large-scale vision language training with bounding box fine-tuning on object detection datasets (Gupta et al., 2019; Lin et al., 2014). Recently, the YouCook-Interactions dataset (Tan et al., 2021) and CoMMA (Tan et al., 2021) have been proposed for the spatial grounding of objects and actions with multimodal self-supervision from HowTo100M videos. All these works assume that the video is temporally clipped with respect to the grounding phrase.

Compared to that, **Temporal Video Grounding** aims to determine the set of consecutive frames corresponding to a text query in an *untrimmed* video (Jiang et al., 2014; Soldan et al., 2021), thus predicting temporal boundaries of action instances. Recent work such as MIL-NCE (Miech et al., 2020), MCN (Chen et al., 2021), and VideoCLIP (Li et al., 2022b) utilize large-scale pretraining for grounding actions temporally via text-to-frame similarity matching on video datasets such as Min-ingYouTube (Kuehne et al., 2019) or CrossTask (Zhukov et al., 2019) without proposals. However, the majority of methods lack spatial localization ability (Zeng et al., 2020; Zhao et al., 2021).

---

[1]We will make the code and the annotations publicly available.

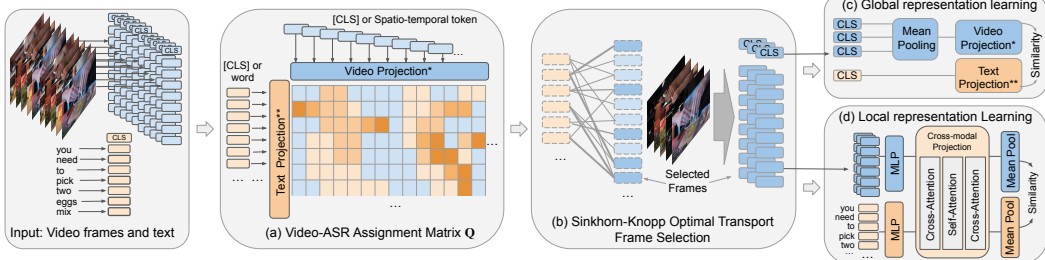

Figure 2: **Spatio-temporal grounding approach.** (a) We aim to select frames with groundable objects and actions. To this end, projected text features are matched with respective frame features. (b) Sinkhorn optimal transport is then leveraged to optimize the selected frames wrt. the text input. (c) Based on the selected frames, a global representation is learned to allow for temporal localization, as well as (d) a local representation to ground the action description in the spatial region.

## 3 A GLOBAL-LOCAL FRAMEWORK FOR SPATIO-TEMPORAL GROUNDING

### 3.1 GENERAL SETUP

The goal of the proposed method is to temporally and spatially localize actions based on free-text queries in untrimmed videos. To this end, two representations are learned, a local and a global one. We start with narrated video clips, each associated with a corresponding visual representation and text narration. Namely, for each clip $\mathcal{X} = \{\mathcal{V}, \mathcal{S}\}$, let $\mathcal{V}$ stand for the video clip and $\mathcal{S}$ for the text narration sentence generated by the automatic speech recognition (ASR) system. Each clip $\mathcal{V}$ consists of $T \times N$ spatio-temporal tokens $\{v_{t,n}\}$, where $t \in \{1, ..., T\}$ represents the number of frames in the video and $n \in \{1, ..., N\}$ represents the number of spatial grid region tokens or features in a frame. The text sentence $\mathcal{S}$ consists of $K$ words $\{s_1, ..., s_K\}$. We represent localized features by the tokens from each modality, and the global features $\{V, S\}$ are acquired either by mean-pooling over the local features or by using the CLS token from the transformer as in Radford et al. (2021). We learn transformations $f : V \to \mathbb{R}^d$ to a $d$-dimensional representation $f(V) \in \mathbb{R}^d$ from the global representation $V$, and $g : S \to \mathbb{R}^d$, to produce similar $d$-dimensional text global embeddings: $g(S) \in \mathbb{R}^d$. Similar to $\{f, g\}$, we note $\{f', g'\}$ to be the transform for localized features, where local features $\{v, s\}$ are also projected as d-dimensional representations.

### 3.2 REPRESENTATION GUIDED FRAME SAMPLING

Learning from multimodal self-supervision is challenging since the narration is likely to be noisy, thus containing more information than the actual task descriptions due to poor temporal alignment or cut scenes (Han et al., 2022), which is the key differences between weakly supervised vision-caption grounding and multimodal self-supervised grounding. This work pursues a frame selection strategy to improve object grounding and temporal alignment during training. We start from a longer sequence $U$, where $U > T$, which includes the video frames before and after the ASR boundaries that could contain actions or objects in the sentence. Our goal is to find $T$ frames out of the $U$ frames that are most relevant to the actions and objects in the sentence $\mathcal{S}$. We formalize it as an optimal transport problem utilizing the Sinkhorn-Knopp algorithm (Cuturi, 2013).

**Optimal transport for text-to-frame assignment.** To acquire the optimal assignment from word features to video frames, an assignment matrix $\mathbf{Q}$ is computed from each video and ASR pair as shown in Figure 2(a). This cross-model optimal transport mechanism is applied to assignment $\mathbf{Q}$ from the projected cross-model similarity $\mathbf{P}$ between word tokens and each video frame, where $\mathbf{P} = g(\mathcal{S}) \bigotimes f(\mathcal{V})^\top \in \mathbb{R}^{K \times U}$. To compute the assignment matrix, the text and video projection layers from the global representation in Figure 2(c) are used to project multimodal features into a common space for feature similarity calculation. We investigate various granularities of the features where we compute the similarity between the text features at the word (local) / sentence (global) level and the visual feature at frames (global) / spatiotemporal tokens (local) level to acquire $\mathbf{P}$, as shown in Table 5. To ensure that the word-to-frame assignment contains more diversity instead of just saturated assignments to a single video frame, we add a constraint by Sinkhorn that requires label assignments to be equally distributed across various video frames representing diverse object/action concepts. Details of the Sinkhorn optimal transport are included in the appendix 7.1.

### 3.3 LOCAL REPRESENTATIONS FOR SPATIAL LOCALIZATION

To capture multimodal interaction with finer granularity, we need to learn the projection between tokenized features as shown in Figure 2(d). We extract spatio-temporal region features $v_{tn}$ from the video. Also, we extract word features $s_k$ which represents the feature from word $k$. All tokenized features are projected through a linear layer. To compute attention between the tokenized features, we stacked two cross-modal attention layers with a self-attention layer in the middle, as illustrated in Figure 2 (d). Cross-modal attention is computed similar to the standard attention mechanism (Lee et al., 2018). Given a spatio-temporal token $v_{tn}$ from a video, we compute the attention score to all of the words $s_k$, where $k \in \{1, ..., K\}$ in the ASR sentence $\mathcal{S}$ by $\alpha_{tnk} = \frac{\exp(e_{tnk})}{\sum_{k=1}^{K} \exp(e_{tnk})}$ in the same video clip, where $e_{tnk} = cosine(v_{tn}, s_k)$. We then acquire a contextual video token feature $\bar{v}_{tn} = \sum_{k=1}^{K} \alpha_{tnk} s_k$, which encoded text contextual information. Note that the contextual vector is represented by aggregating the representations from the other modality. Follow the standard self-attention computation (Vaswani et al., 2017) $K, Q, V$ represent the features for the keys, queries, and values as: $Attn(K, Q, V) = \text{softmax}\left(\frac{(Q^\top K)}{\sqrt{d_k}}\right) V$ where $d_k$ is the dimension of the key. In our case, we feed each contextual features $\{\bar{v}_{tn}, \bar{s}_k\}$ right after the first cross-attention layer to be the $K, Q, V$ to acquire its self-attended representation. The localized attention model was trained using contrastive loss. To represent the video clip $\mathcal{V}$ and ASR sentence $\mathcal{S}$, we mean-pool over the spatio-temporal tokens in video $\bar{V} = \frac{1}{TN} \sum_{r=1}^{TN} \bar{v}_r$, and words $\bar{S} = \frac{1}{K} \sum_{k=1}^{K} \bar{s}_k$ respectively. Let $\left(\bar{V}^{(l)}, \bar{S}^{(l)}\right)$ be the $l$-th training example pair. We adopt the Noise Contrastive Estimation (NCE) loss (Gutmann & Hyvärinen, 2010) and the localized attention losses $\mathcal{L}_{Local}$ :

$$-\frac{1}{B} \sum_{l=1}^{B} \left[ \left( \log \frac{e^{\bar{V}_l \cdot \bar{S}_l - \delta}}{e^{\bar{V}_l \cdot \bar{S}_l - \delta} + \sum_{\substack{k=1 \\ k \neq l}}^{B} e^{\bar{V}_k^{imp} \cdot \bar{S}_l}} \right) + \left( \log \frac{e^{\bar{V}_l \cdot \bar{S}_l - \delta}}{e^{\bar{V}_l \cdot \bar{S}_l - \delta} + \sum_{\substack{k=1 \\ k \neq l}}^{B} e^{\bar{V}_l \cdot \bar{S}_k^{imp}})} \right) \right] \quad (1)$$

where $B$ is the batch. $\bar{V}_k^{imp}$ and $\bar{S}_k^{imp}$ represent imposter samples, and $\delta$ is a margin hyperparameter.

### 3.4 GLOBAL REPRESENTATIONS FOR TEMPORAL LOCALIZATION

We learn the global representation of a video clip and a sentence by contrastive loss, as shown in Figure 2(c). We again use the NCE loss function (Gutmann & Hyvärinen, 2010). The global contrastive loss $\mathcal{L}_{Global}$ follows the formulation as Equation 1 while using the global representations $V$ and $S$, which are the [CLS] tokens from both modalities, instead of the local representations. Projecting the global features to the same space ensures that the features across different modalities are comparable. Since global representations encode information from the entire video, it is essential in encoding temporal information for the later downstream tasks. The final model is optimized by the sum of both losses as $\mathcal{L}_{Final} = \mathcal{L}_{Local} + \mathcal{L}_{Global}$.

### 3.5 INFERENCE FOR SPATIO-TEMPORAL GROUNDING.

To perform spatio-temporal grounding on untrimmed videos, we start from temporal action detection as shown in Figure 3. Given a pool of possible action descriptions on the left and an untrimmed video, we perform feature similarity matching using the global representation ([CLS] token) per frame with a threshold $\tau$ to filter backgrounds. We pick the action class with the largest similarity score per frame. Later, we use the predicted action class and feed it into the local representation branch to compute spatial grounding. We follow attention rollout (Abnar & Zuidema, 2020) to compute feature similarity between visual tokens and text to-

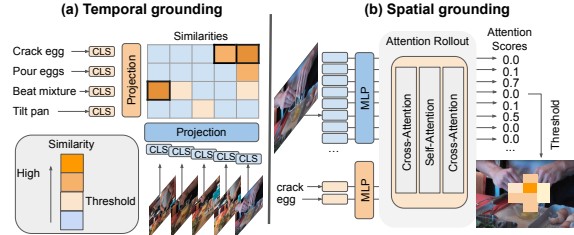

Figure 3: **Spatio-temporal inference.** Both temporal and spatial representations are used for spatio-temporal grounding: Starting by predicting the action boundary, spatial grounding is performed on the selected frames using the predicted label to find corresponding regions.

kens through the cross-attention and self-attention. In the end, we acquire an attention heatmap for later downstream evaluation. More information on inferencing are in appendix 8.2.

# 4 GROUNDINGYOUTUBE BENCHMARK

Current downstream datasets either provide spatial (Tan et al., 2021), temporal annotation (Kuehne et al., 2019), or short video clip with spatio-temporal annotation (Zhang et al., 2020). These datasets do not provide the opportunity to evaluate both aspects, spatial and temporal grounding, in an untrimmed long video manner. We, therefore, extend one of the current benchmarks, MiningYouTube (Kuehne et al., 2019), which already provides dense temporal annotations, and we annotate video clips in the dataset with spatial information.

Annotating the spatio-temporal extent of actions can be challenging as there is no clear visible outline as in object annotation, nor is there a unique signal to indicate the temporal begin and end points. Similarly, grounding systems do not produce pixel-exact bounding boxes, but rather indicate regions of interest. Detector-free spatial grounding models (Arbelle et al., 2021) address this fuzziness by relying on pointing game accuracy, thus only using the center point of the heat map for evaluation. Lending on this idea, annotators were asked to mark the presumed center point of the action. Compared to bounding boxes, center point annotation can be advantageous because annotators are not visually distracted by object outlines, so it is more likely that the most important region will be selected. We capture five annotations per frame, resulting in a density-based heat map.

Starting from 5,091 clips showing one of the 512 action classes, we adopt the methodology used for temporal action localization developed in (Gu et al., 2018) and label one frame per second, resulting in $26,987$ frames. We annotated all frames with five repeats per image, resulting in five annotations per frame and $134,935$ point labels in total. Following the previous evaluation setting using bounding boxes (Kalogeiton et al., 2017), we get the union of all annotated points in a single frame with an additional distance to construct the bounding box. We provide more information on the annotation process and dataset analysis are in the appendix 10. Video examples are in the supplement.

# 5 EXPERIMENTS

## 5.1 DATASETS

**Training Data: HowTo100M dataset** contains 1.2M instructional videos along with their corresponding automatically generated speech (ASR) transcriptions.

**Downstream Datasets: GroundingYoutube (GYT)** is used to evaluate the task of multi-action spatio-temporal grounding as described in Section 4. **MiningYoutube (MYT)** (Kuehne et al., 2019) provides temporal annotation and is limited to the domain of cooking instruction videos. **YouCook-Interaction (YC-Inter)** (Tan et al., 2021) is an extension of the YouCook2 dataset (Zhou et al., 2018b) for cooking instruction providing bounding boxes for 6K selected frames. The bounding boxes usually comprise the hand and the tool mentioned in the respective sentence-wise annotation. To further benchmark on general video domains on the **V-HICO** dataset (Li et al., 2021) with 6.5k videos with human-object interaction bounding boxes annotations, and **Daly** action dataset (Weinzaepfel et al., 2016), featuring videos consisting of daily actions such as "brushing teeth".

## 5.2 BASELINE METHODS

The proposed system is compared to various multimodal methods based on self- and weak supervision: MIL-NCE (Miech et al., 2020), which utilizes S3D (Xie et al., 2018) and word2vec (Mikolov et al., 2013) to project two modalities into a common space, is chosen as the standard baseline for this task; CoMMA (Tan et al., 2021), the best-performing model for spatial representations in self-supervised learning (we denote CoMMA† to represent the model that uses weights shared by the author[2]); CLIP (Radford et al., 2021), an image-text model trained with transformer architecture, is further applied as the backbone and trained with (Tan et al., 2021) to construct CoMMA‡;

---

[2]We thank the authors for providing code and weights.

| Method | Backbone | DataSet | Supervision | Modality | GroundingYoutube | | | | | | |
|---|---|---|---|---|---|---|---|---|---|---|---|
| | | | | | IoU+Point | mAP | | | | | |
| | | | | | | 0.1 | 0.2 | 0.3 | 0.4 | 0.5 | 0.1:0.5 |
| CoMMA† (Tan et al., 2021) | S3D | HT250K | Self | VT | 1.02 | 2.18 | 1.72 | 1.11 | 0.93 | 0.37 | 1.26 |
| MIL-NCE (Miech et al., 2020) | S3D* | HT100M | Self | VT | 4.67 | 33.94 | 25.16 | 12.65 | 3.42 | 0.41 | 15.11 |
| Ours | S3D | HT100M | Self | VT | **9.12** | **42.70** | **35.49** | **25.16** | **16.22** | **10.05** | **25.92** |
| GLIP (Li et al., 2022a) | Swin-L* | Cap24M | Weak | IT | 1.24 | 2.83 | 2.10 | 1.52 | 0.96 | 0.37 | 1.56 |
| CoMMA‡ (Tan et al., 2021) | CLIP | HT100M | Self | VT | 1.68 | 3.51 | 2.32 | 1.88 | 0.99 | 0.40 | 1.82 |
| CLIP (Radford et al., 2021) | CLIP | HT100M | Self | IT | 3.59 | 29.54 | 22.15 | 9.16 | 2.48 | 0.39 | 12.74 |
| RegionCLIP (Zhong et al., 2022) | ResNet-101* | CC3M | Weak | IT | 5.65 | 35.65 | 27.43 | 15.69 | 4.31 | 0.86 | 16.78 |
| Ours | CLIP | HT100M | Self | VT | 10.09 | 42.81 | 36.05 | 25.84 | 17.10 | 11.35 | 26.63 |
| Ours | CLIP* | HT100M | Self | VT | **11.53** | **43.64** | **36.94** | **26.78** | **19.45** | **14.61** | **28.26** |
| MIL-NCE(temp.)+RegionCLIP(spa.) | - | - | - | VT | 9.21 | 40.54 | 34.97 | 22.38 | 13.79 | 9.18 | 22.33 |

Table 2: **Spatio-temporal grounding on GroundingYouTube full videos**. (V: video, I: image, T: text.) * indicates finetuned backbone.

GLIP (Li et al., 2022a) and RegionCLIP (Zhong et al., 2022), state-of-the-art image-text grounding models that combine large-scale image caption pretraining and object detection fine-tuning, which we consider weakly supervised as the bounding box proposal network was trained on other human-annotated data. We further construct a strong baseline MIL-NCE+RegionCLIP where we use MIL-NCE for temporal localization and RegionCLIP for spatial grounding following the inference pipeline of Figure 3 without additional training. Finally, two fully supervised models, Tube-DETR (Yang et al., 2022a) and STCAT (Jin et al., 2022) are included for comparison. Further details of the implementation and experimental settings can be found in the appendix 8.1. Inference setups for each baseline are described in Section 8.2.

## 5.3 DOWNSTREAM TASKS

We considered the following downstream tasks to evaluate spatio-temporal grounding abilities of various models (detailed description is included in the appendix 8.3):

(i) **Spatio-temporal grounding in untrimmed video** is evaluated on the proposed Grounding Youtube dataset. The entire video and the respective pool of action instructions were provided. The model needs to localize each action step in time (start-time/end-time) and space (location in the video) as described in Figure 3. We evaluate in two metrics: **IoU+Pointing game** combines spatial grounding (Akbari et al., 2019) and temporal grounding (Kuehne et al., 2019) metrics. We also compute **video mAP** following previous evaluation (Gu et al., 2018), where we set IoU threshold between GT and predicted spatio-temporal tubes. A prediction is correct when it surpasses the IoU threshold. We compute the mAP over all classes.

(ii) **Spatial grounding** is given a text description to localize the corresponding region in the trimmed video. This task is evaluated using the **pointing game accuracy**. If the predicted point lies in the ground truth bounding box, the result counts as a "hit" and counts as "miss" otherwise. The final accuracy is calculated as a ratio between hits to the total number of predictions $\frac{\# \text{hits}}{\# \text{hits} + \# \text{misses}}$. We also report the mean average precision (**mAP**) following the settings from V-HICO (Li et al., 2021).

(iii) **Temporal grounding** provides videos with the respective actions and their ordering, including the background. The goal is to find the correct frame-wise segmentation of the video. We follow the inference procedure in (Kuehne et al., 2019) to compute the alignment given the similarity input matrix. The task is evaluated by intersection over detection (IoD), defined as $\frac{G \cap D}{D}$ the ratio between the intersection of ground-truth action $G$ and prediction $D$ to prediction $D$, and the Jaccard index, which is an (IoU) given as $\frac{G \cap D}{G \cup D}$.

## 5.4 COMPARISON WITH STATE-OF-THE-ART METHODS

(i) **Spatio-temporal grounding in untrimmed video:** We first compare the proposed method with other approaches designed for spatial or temporal grounding in Table 2. It shows that models without specific loss designs for spatial grounding (MIL-NCE (Miech et al., 2020), CLIP (Radford et al., 2021)) show good mAP scores but lower pointing game accuracy. Out of the two weakly supervised methods, GLIP (Li et al., 2022a) and RegionCLIP (Zhong et al., 2022)), trained with aligned image-text, RegionCLIP show significantly better performance in this setting, while both perform

| | | | | | YC-Inter | GroundingYT | | V-HICO | | Daly | |
|---|---|---|---|---|---|---|---|---|---|---|---|
| Method | Backbone | Data | Super. | Mod. | Acc | Acc | mAP | Acc | mAP | Acc | mAP |
| MIL-NCE (Miech et al., 2020) | S3D* | HT100M | Self | VT | 23.67 | 27.45 | 8.21 | 12.65 | 11.23 | 13.84 | 24.23 |
| CoMMA† (Tan et al., 2021) | S3D | HT250K | Self | VT | 48.63 | 47.68 | 23.38 | 40.97 | 21.45 | 54.48 | 33.39 |
| Ours | S3D | HT100M | Self | VT | **53.98** | **60.62** | **44.93** | **44.32** | **24.31** | **66.35** | **45.93** |
| CLIP (Radford et al., 2021) | CLIP | HT100M | Self | IT | 14.10 | 12.50 | 3.49 | 29.23 | 12.51 | 18.02 | 27.28 |
| CoMMA‡ (Tan et al., 2021) | CLIP | HT100M | Self | VT | 52.65 | 47.56 | 36.42 | 55.20 | 34.54 | 61.06 | 44.37 |
| RegionCLIP (Zhong et al., 2022) | RN50x4* | CC3M | Weak | IT | 51.56 | 52.84 | 23.42 | 57.92 | 37.82 | 67.12 | 48.62 |
| GLIP (Li et al., 2022a) | Swin-L* | Cap24M | Weak | IT | 52.84 | 53.62 | 24.73 | **66.05** | 41.17 | - | - |
| Ours | CLIP | HT100M | Self | VT | 57.10 | 55.49 | 43.12 | 60.71 | 39.28 | 70.08 | 50.56 |
| Ours | CLIP* | HT100M | Self | VT | **58.35** | **56.98** | **45.32** | 62.34 | **41.56** | **71.35** | **52.78** |
| TubeDETR (Yang et al., 2022a) | MDETR | Vid-STG | Full | VT | 51.63 | 53.24 | 41.76 | 63.23 | 40.87 | 84.21 | 62.98 |
| STCAT (Jin et al., 2022) | ResNet-101 | Vid-STG | Full | VT | 54.47 | 55.90 | 44.21 | 65.34 | 41.10 | 85.42 | 63.94 |

Table 3: **Video spatial grounding**. We evaluate the accuracy of the pointing game and the mean average precision. * indicates finetuned backbone.

| | GroundingYT Spatio-temporal | MiningYT Temporal | YC-Inter. Spatial | GroundingYT Spatio-temporal | MiningYT Temporal | YC-Inter. Spatial |
|---|---|---|---|---|---|---|
| | | w/o Sinkhorn | | | with Sinkhorn | |
| None | 17.43 | 18.34 | 57.42 | - | - | - |
| Global T - Global V | 18.24 | 19.38 | 56.31 | 18.96 | 19.89 | 57.34 |
| Global T - Local V | 18.01 | 19.31 | 57.56 | 18.53 | 19.35 | 57.67 |
| Local T - Global V | 18.05 | 19.85 | 57.34 | 19.32 | **20.36** | 58.13 |
| Local T - Local V | 18.31 | 19.48 | 57.86 | 19.43 | 20.16 | **58.51** |
| Average over last two | 18.36 | 19.68 | 57.77 | **19.45** | 20.33 | 58.35 |

Table 5: **Frame selection: (a)** Sinkhorn selection results in better supervision. **(b)** We investigate all possible combinations of global and local representations for frame selection similarity matching.

in a similar range in the spatial grounding scenario (see Table 3). We attribute this behavior to the fact that RegionCLIP distinguishes frames with relevant queries better from background than GLIP, leading to better temporal localization. Further, the proposed method improves over the other baselines underlining the need to incorporate global (temporal) and local (spatial) representations. We compared our strong baseline MIL-NCE+RegionCLIP, which combines two approaches specialized in temporal and spatial aspects, to our task. Experiments showed that combining a joint objective that learns spatial and temporal information jointly results in better performance than simply applying the best temporal and spatial model. Also, such a combine objective also benefits more when the visual backbone is finetued as well. We construct a split with single action shown in appendix 9.2.

(ii) **Spatial grounding:** Second, we compare the performance of the proposed framework to other methods on the task of spatial grounding, including models with weak supervision, as well as models trained in a fully supervised setting in Table 3. In the instruction video domain (GYT and YC-Inter), the proposed approach achieves the best result among all weakly and self-supervised trained methods. In the general domain (V-HICO and Daly), the method also achieves competitive results, showing the generalizability of the model to other domains. Note that in the Daly dataset, the classes are verbs, which are not detectable by the object-focused model GLIP. Compared to their weakly trained counterparts, fully-supervised model (TubeDETER (Yang et al., 2022a), STCAT (Jin et al., 2022)) achieve competitive performance in the general domain (V-HICO, Daly) and slightly lower performance in instruction domain (GYT, YC-Inter) due to the domain gap with respect to the training data.

(iii) **Temporal grounding:** We finally evaluate temporal grounding in Table 4. Here, it shows that global representations also profit from local representation learning. This hypothesis is further validated in the ablation studies in Table 6, where we ablate both losses for all three settings and show a consistent improvement in the joint loss formulation.

| Method | Backbone | Data | Super. | IoU | IoD |
|---|---|---|---|---|---|
| Mining: MLP (Miech et al., 2020) | TSM | MiningYT | Weak | 9.80 | 19.20 |
| CoMMA* (Tan et al., 2021) | S3D | HT250K | Self | 2.05 | 5.63 |
| MIL-NCE (Miech et al., 2020) | S3D* | HT100M | Self | 18.69 | 26.74 |
| Ours | S3D | HT100M | Self | 19.18 | 27.65 |
| Ours | CLIP | HT100M | Self | 19.88 | 28.50 |
| Ours | CLIP* | HT100M | Self | **20.33** | **29.67** |

Table 4: **Temporal Grounding on MiningYoutube.** * indicates finetuned backbone.

Figure 4: **Visualization on GroundingYoutube.** Red box: annotation. Heatmap: prediction.

## 5.5 ABLATION STUDY

We perform ablation studies with respect to all three settings, spatio-temporal grounding, as well as spatial and temporal grounding alone, reporting performance for spatio-temporal grounding on GroundingYT using mAP with IoU@0.4, on temporal grounding using MiningYT IoU, and on spatial grounding using YC-Inter. pointing game. Additional ablation are in appendix 9.3.

**Frame selection strategy.** We perform an ablation on the possible frame selection strategies for our method (Figure 2(b) and Section 3.2). In Table 5, *None* uses all frames within the ASR boundary ($U = T$) as our video training data. *Global* represents the [CLS] token in text and video. *Local* uses the words and spatio-temporal tokens. In the setting Sinkhorn was not applied, the top $T$ frames with the highest similarity score were selected. When we set spatio-temporal tokens as the selection target, we sum over the scores with respect to each frame to acquire the frame similarity score. It shows that selecting frames based on Sinkhorn selection leads to consistently better results as it enforces more variety of visual concepts but also captures frames with possible groundable objects. It further shows that word tokens are more suitable than the global text CLS token for frame selection. Finally, we see that depending on the task (spatial vs. temporal), a local resp. global representation is better, and a combination of both works best for spatio-temporal grounding. We provide runtime analysis of such frame selection strategy in the appendix 9.1.

**Global and local loss.** As mentioned in the spatio-temporal evaluation, both features contribute significantly to the final grounding result. We test the model by ablating out each loss. As shown in Table 6, not only that each loss contributes to the task of spatio-temporal grounding on the GYT, but also the whole is more than the sum of its parts (losses) since this task requires both spatial and temporal detection. The re-

|  | GroundingYT Spatio-temp | MiningYT Temporal | YC-Inter. Spatial |
|---|---|---|---|
| only Local loss | 7.29 | 5.23 | 55.29 |
| only Global loss | 9.28 | 19.12 | 36.23 |
| w/ Both loss | 19.45 | 20.33 | 58.35 |

Table 6: **Loss ablations:** both losses contribute to the final loss, and the existence of global loss helps the localization task.

duced impact of the global loss in the case of YC-Inter is based on the fact that this is a pure spatial grounding dataset (no background frames) without temporal detection, and the local loss plays a more critical role. We observe the same patterns in the temporal grounding result for MYT, where spatial localization wasn't directly contributing to the final performance.

## 5.6 QUALITATIVE RESULTS

We visualize our spatio-temporal result in Figure 4. For the GLIP model, we output the bounding box with the highest confidence score and visualize its center point. We found GLIP model focuses on the salient object while our model focuses more on human-object interaction.

## 6 CONCLUSION

We presented an approach for learning spatio-temporal grounding with self-supervision and a new dataset: GroundingYoutube annotations, where we densely annotate spatio-temporal points/boxes from untrimmed multi-action videos. Our method includes a frame selection mechanism that identifies frames with groundable objects to adapt the learning process for untrimmed videos. Furthermore, we jointly learn global representations, which capture temporal information, and local representations learning fine-grained multimodal interactions between video and text. We conducted extensive experiments to evaluate the performance of our approach showing state-of-the-art performance in spatio-temporal grounding, as well as temporal and spatial grounding alone.

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

**This appendix is organized as follows:**

# 7 METHOD DETAILS

## 7.1 SINKHORN OPTIMAL TRANSPORT

To acquire the optimal assignment from word features to video frames, an assignment matrix $\mathbf{Q}$ is computed from each video and ASR pair as shown in Figure 2(a). This cross-model optimal transport mechanism is applied to assignment $\mathbf{Q}$ from the projected cross-model similarity $\mathbf{P}$ between word tokens and each video frame, where $\mathbf{P} = g(\mathcal{S}) \bigotimes f(\mathcal{V})^{\top} \in \mathbb{R}^{K \times U}$. To compute the assignment matrix, the text and video projection layers from the global representation in Figure 2(c) are used to project multimodal features into a common space for feature similarity calculation. To ensure that the word-to-frame assignment contains more diversity instead of just saturated assignments to a single video frame, we add a constraint that requires label assignments to be equally distributed across various video frames representing diverse object/action concepts. This is achieved by restricting $\mathbf{Q_v}$ to a *transportation polytope* $\mathcal{Q}_v$:

$$\mathcal{Q} = \left\{ \mathbf{Q} \in \mathbb{R}_{+}^{U \times K} \mid \mathbf{Q}\mathbf{1}_K = \tfrac{1}{U}\mathbf{1}_U, \mathbf{Q}^{\top}\mathbf{1}_U = \tfrac{1}{K}\mathbf{1}_K \right\}, \tag{2}$$

which enforces the soft-assignment distribution $\mathbf{Q}$ to assign an equal marginal probability to each of the $U$ frames instead of converging to a single frame. The vector $\mathbf{1}_U$ represents one vector with dimension $U \times 1$.

The next goal is to enforce this *transportation polytope* $\mathcal{Q}$. A solution for $\mathbf{Q}$ is now computed using the optimal transport Sinkhorn-Knopp algorithm (Caron et al., 2020; Cuturi, 2013) as shown in Figure 2(b). The Sinkhorn-Knopp algorithm also normalizes the distribution of $\mathbf{P}$ as:

$$\mathbf{Q} = \text{Diag}(\alpha) \exp\left(\tfrac{\mathbf{P}}{\varepsilon}\right) \text{Diag}(\beta), \tag{3}$$

where $\alpha$ and $\beta$ are scaling vectors that restrict $\mathbf{Q}$ to have a uniform distribution across region assignment. $\varepsilon$ is a parameter that controls the smoothness of the mapping (Caron et al., 2020).

The $T$ frames are then selected by the corresponding assignment $\mathbf{Q}$ from the frames with top $T$ aggregated similarity sum over each word for further training. Note that the selection part $\mathbf{P}$ is from a trainable projection. While acquiring a better word-to-region projection during training, we hypothesize that the frame selection also benefits. The respective frame selection strategy is evaluated in Table 5.

# 8 EXPERIMENTAL SETUP

## 8.1 BACKBONES AND TRAINING

We evaluate the proposed method on backbones, CLIP (Radford et al., 2021) and S3D-word2vec (Miech et al., 2020). We described the detailed setup as well as the training in the following.

**CLIP models.** For both the visual and text backbone, we use the pretrained weights from CLIP (Radford et al., 2021) with transformer ViT-B/32 and fix the encoder. Both the visual and text encoder has a final embedding size of 512. We apply them to video segments with 12-28 seconds, processing 1 frame per second. An evaluation of how many frames to process (identical to the number of seconds) is shown in Table 9. It shows the best results when we start with 80 possible frames $U$ (as described in Section 3.2), from which $T = 16$ frames are selected for training. Ablation of the number of frames $T$ used for training is shown in Table 10. We used a batch size of $B = 64$ video clips.

**S3D-word2vec models.** For the video backbone, we follow (Tan et al., 2021) and use S3D initialized by MIL-NCE on HowTo100M (Miech et al., 2020) at the rate of 5 frames per second and fix the video encoder. The global video clip features were max-pooled over time and projected into embeddings of dimension 512. We used the mean-pooled S3D spatio-temporal features to represent the global representation of the video following the S3D architecture (Xie et al., 2018). For the text feature, we follow (Miech et al., 2019) using a GoogleNews pre-trained word2vec model (Mikolov et al., 2013) and max-pooling over words in a given sentence to acquire the text global feature. We follow (Miech et al., 2020) to use the max-pooled word embedding to represent the sentence (global representation) since there is no [CLS] token. Also, the sentence feature is used for the query word selection instead of the [CLS] token. We use a batch size of $B = 96$ video clips.

**Training.** For the training of both backbone settings, we use an Adam optimizer (Kingma & Ba, 2015) with a learning rate of $1e{-}4$. In the setting of fintining CLIP, we set a learning rate of $1e{-}7$ for the CLIP backbone. The model is trained for 10 epochs on 4 V100 GPUs, which takes about two days.

## 8.2 INFERENCE

**Inference for the proposed model and CoMMA.** For inference in the case of temporal grounding, as shown in Figure 3(a), we first normalize the global feature for video and text. We used a (temporal) threshold $\theta = 0.5$ to separate detections from the background. In spatial grounding, we acquire an attention heatmap using the attention rollout (Abnar & Zuidema, 2020) described in Section 3.5. We set a spatial threshold $\tau = 0.01$ to create the mask, as shown in Figure 3(b). The choice of this spatial threshold is evaluated in Table 12.

**GLIP, RegionCLIP baseline inference.** In spatial grounding, we are given a text query and need to localize it in the frame. GLIP and RegionCLIP predict multiple bounding boxes corresponding to the text query. We select the predicted bounding box with the highest confidence score as the prediction result. We use the center point of the predicted bounding box for the pointing game evaluation as the model prediction. For *mAP* evaluation, we use the predicted bounding box to compute IoU with the ground truth bounding box. In spatio-temporal grounding, we input all possible action description labels as candidates similar to Figure 3(a). We pick the class with the highest confidence score as the predicted label. If the model made no prediction, we would predict it as "background". The spatial inference is the same as the spatial grounding setting.

**TubeDETR, STCAT baseline inference.** TubeDETR and STCAT are spatio-temporal grounding models trained to predict a single spatio-temporal tube per video. In both cases, TubeDETR and STCAT, we use models trained on the Vid-STG dataset with 448x448 resolution and evaluate them for the task of spatial grounding. Since this dataset contains mostly short videos ($<$30sec), we observed that both methods will also only predict a trajectory tube in this temporal range ($<$30sec), no matter how long the input video is. To allow us to apply them to longer videos ($>$30sec), we split the longer videos based on sliding windows of 5-sec for better performance.

**MIL-NCE, CLIP baseline inference.** Both models are trained based on global representations for both input modalities, videos/images and text. We can, therefore, directly compute a sentence-to-video-frame similarity to perform the temporal grounding for Figure 3(a), following the same process as the proposed method for temporal grounding. For spatial grounding, we compute sentence-to-region feature similarity. Both visual backbones produce a 7x7 grid feature. We normalize the sentence and region features, then select a spatial threshold $\tau = 0.5$ to create the mask for the *mAP* evaluation.

## 8.3 EVALUATION METRICS

(i) **Spatio-temporal grounding in untrimmed video** is evaluated on our annotated GroundingYoutube dataset. We combined the spatial and temporal grounding evaluation as before (Kuehne et al., 2019; Akbari et al., 2019) to form the spatio-temporal evaluation. The entire video and the respective pool of action instructions were provided. The model needs to localize each action step in temporal (start-time/end-time) and spatial (location in the video) as described in Figure 3. We evaluate in two metrics: **IoU+Pointing game** combines the evaluation setting from the spatial grounding (Akbari et al., 2019) and temporal grounding (Kuehne et al., 2019) metrics. For each video frame, the pre-

diction is correct when the model predicts the correct action for the frame. Also, given the predicted action as a query, the maximum point of the heatmap aims to lie within the desired bounding box. We then compute the Intersection over Union (IoU) over all the predictions with the GT to acquire the final score. We also compute **video mAP** following previous evaluation (Gu et al., 2018), where we set IoU threshold between GT and predicted spatio-temporal tubes. A prediction is correct when it surpasses the IoU threshold. We then compute the mAP over all classes. We form a 3D prediction mask following Figure 3 and compute IoU between our 3D heatmap and 3D tube.

(ii) **Spatial grounding** is given a text query description to localize the corresponding region in the trimmed video. We use GroundingYoutube, Youcook-Interaction, V-HICO, and Daly for evaluation. This task is evaluated using the **pointing game accuracy**. Given the query text and video, we compute the attention heatmap on the video as described in Figure 3(b). If the highest attention similarity score lies in the ground truth bounding box, the result counts as a "hit" and counts as "miss" otherwise. The final accuracy is calculated as a ratio between hits to the total number of predictions $\frac{\#\,\text{hits}}{\#\,\text{hits}+\#\,\text{misses}}$. We report the mean average precision **(mAP)** following the settings from V-HICO (Li et al., 2021). Given a human-object category as the text query, we aim to localize the spatial location in the video frame. The predicted location is correct if their Intersection over-Union (IoU) with ground truth bounding boxes is larger than 0.3. Since we do not use any bounding box proposal tools or supervision, we create an attention heatmap as described in Figure 3(b) to create a mask for IoU computation. We follow (Li et al., 2021) and compute the mAP over all verb-object classes.

(iii) **Temporal grounding** provides videos with the respective actions and their ordering, including the background. The goal is to find the correct frame-wise segmentation of the video. We follow the inference procedure in (Kuehne et al., 2019) to compute the alignment given our similarity input matrix. The task is evaluated by intersection over detection (IoD), defined as $\frac{G \cap D}{D}$ the ratio between the intersection of ground-truth action $G$ and prediction $D$ to prediction $D$, and the Jaccard index, which is an (IoU) given as $\frac{G \cap D}{G \cup D}$.

# 9 ADDITIONAL EXPERIMENTS

## 9.1 RUNTIME ANALYSIS

We analyze the computational costs of sampling and loss. We sample 16-second videos at a frame rate of 5 FPS (80 frames in total). We report the execution time for a single batch (batch size = 64) averaged over 100 batches. For the *frame sampling strategy*: (1) Random select 8 frames: 1.48s. (2) Optimal transport based selection of 8 frames out of 64: 1.54s. (3) Entire 64 frames: 1.74s. The execution time of our method is close to traditional random sampling while capturing diverse visual concepts, which improves the training process. For the *global and local* components: (1) Global loss only: 1.1s. (2) Local loss only: 1.52s. (3) Both losses: 1.54s. Computation of the local loss is more time-consuming than the global loss due to its requirement for features with finer granularity.

## 9.2 SINGLE-ACTION SPATIO-TEMPORAL GROUNDING.

Current spatio-temporal detection and grounding datasets (Jiang et al., 2014; Gu et al., 2018) usually aim to discriminate a single given action class from the background class in a short clip. This differs from our setup of spatio-temporal grounding in untrimmed videos, which usually comprises a set of phrases that need to be detected in a 3-5 min long video. To allow an evaluation of spatio-temporal grounding approaches based on single phrase grounding, we construct a clip-level evaluation where the clip varies from 9 sec to 60 sec. Given an action step, we append the video segments before and after the steps with the same time length of the action step to form the final video clip. This results in 2,895 clips for the spatio-temporal clip grounding evaluation. For each clip, the temporal action intervals occupy 33% of corresponding videos, which demonstrates the difficulty of the setting. In this setting, instead of selecting the possible action step from a pool, the ground truth action step was given as the text query for spatio-temporal grounding. This allows us to directly compare with supervised spatio-temporal grounding methods (Yang et al., 2022a; Jin et al., 2022) as described in Section 5.4. As shown in Table 7, we observe that the baseline GLIP models achieve a much better performance compared to Table 2. This is due to the fact that this setting does not require the model to select the text query from the pool, which the GLIP model was not trained to do. Moreover, we

| Method | Backbone | DataSet | Supervision | Modality | IoU+Point | 0.1 | 0.2 | 0.3 | 0.4 | 0.5 | 0.1:0.5 |
|---|---|---|---|---|---|---|---|---|---|---|---|
| | | | | | | | | GroundingYoutube mAP | | | |
| CoMMA* (Tan et al., 2021) | S3D-word2vec | HT250K | Self | VT | 1.10 | 7.46 | 5.84 | 4.20 | 2.65 | 1.53 | 4.93 |
| MIL-NCE (Miech et al., 2020) | S3D-word2vec | HT100M | Self | VT | 12.41 | 45.91 | 32.33 | 15.35 | 3.70 | 2.56 | 19.54 |
| Ours | S3D-word2vec | HT200K | Self | VT | 19.46 | 51.95 | 40.31 | 26.81 | 16.27 | 7.81 | 28.63 |
| CoMMA† (Tan et al., 2021) | CLIP | HT200M | Self | VT | 2.64 | 8.94 | 6.89 | 5.47 | 4.18 | 2.67 | 5.63 |
| CLIP (Radford et al., 2021) | CLIP | HT200K | Self | IT | 11.34 | 43.28 | 30.64 | 11.20 | 3.10 | 1.94 | 18.03 |
| RegionCLIP (Zhong et al., 2022) | ResNet-101 | CC3M | Weak | IT | 17.42 | 51.86 | 40.23 | 26.10 | 15.23 | 7.29 | 28.14 |
| GLIP (Li et al., 2022a) | Swin-L | Cap24M | Weak | IT | 18.15 | 52.61 | 41.83 | 26.93 | 17.23 | 8.46 | 29.41 |
| Ours | CLIP | HT200K | Self | VT | 20.81 | 53.24 | 42.96 | 29.17 | 20.36 | 11.84 | 31.51 |
| TubeDETR (Yang et al., 2022a) | MDETR | Vid-STG | Full | VT | 26.43 | 63.47 | 50.95 | 38.23 | 28.31 | 19.34 | 40.06 |
| STCAT (Jin et al., 2022) | ResNet-101 | Vid-STG | Full | VT | 27.84 | 64.96 | 52.13 | 40.61 | 30.49 | 20.55 | 41.75 |

Table 7: **Single-action spatio-temporal grounding in short videos.** We compare spatio-temporal grounding approaches based on single phrase grounding. To this end, we construct a clip-level evaluation based on the action segments of GroundingYouTube, where each action segment varies from 9 sec to 60 sec. We append video segments before and after the annotated action with the same time length of the action step to form the final video clip. This allows us to directly compare with supervised spatio-temporal grounding methods (Yang et al., 2022a; Jin et al., 2022).

find that weakly supervised methods, GLIP and RegionCLIP, show only limited ability to differentiate the queried action from the background, which leads the model to ground the text query in most of the frames. However, both demonstrate powerful localization ability in foreground action segments, which results in a decent performance. The fully-supervised trained models (TubeDETR, STCAT) achieved a balance in localizing temporally and spatially, resulting in the best performance on this task.

| Attention Architecture | GroundingYT Spatio-temporal | MiningYT Temporal | YouCook-Inter. Spatial |
|---|---|---|---|
| Self+Cross | 15.4 | 18.7 | 54.1 |
| Cross+Self | 15.9 | 18.9 | 54.5 |
| Cross+Cross | 16.5 | 19.3 | 56.2 |
| Cross+Self+Cross | 17.1 | 19.9 | 57.1 |

Table 8: **Ablation on different attention architecture**

| # of frames | 60 | 80 | 100 | 120 | 140 |
|---|---|---|---|---|---|
| **GYT (Spatio-temporal)** | 16.4 | 17.1 | 17.0 | 16.8 | 16.1 |
| **YC-Inter (Spatial)** | 56.3 | 57.1 | 56.8 | 56.7 | 55.9 |

Table 9: **Ablation of # of frames used for selection**

## 9.3 ABLATION AND DECISION CHOICES

We performed additional ablation studies using the CLIP backbone without finetuning.

**Attention architecture.** We tested different architectures by stacking the self-attention or cross-attention block in the model to calculate contextualized local representations, as shown in Figure 2(d). As shown in Table 8, we found that the standard multimodal transformer architecture (self+cross) to have the worst performance. Using two cross-attention blocks was beneficial in incorporating more cross-modal interaction between local features. Finally, including a self-attention layer slightly improves the final representations by encoding better single-modality representations.

**Frames used for selection.** As shown in Table 9, we perform an ablation study on the number of candidates frames $U$ used for training. We found that selecting 80 frames (16 seconds) achieves the best performance, comprising the useful video information in training while not including too many irrelevant concepts that diverge from the action/object in the ASR sentence.

| Frame length | 1 | 4 | 8 | 16 | 24 |
|---|---|---|---|---|---|
| **GYT (Spatio-temporal)** | 5.2 | 9.5 | 16.1 | 17.1 | 16.5 |
| **YC-Inter (Spatial)** | 31.1 | 48.2 | 55.5 | 57.1 | 56.1 |

Table 10: **Effect of # video frames used for training**

| Train/test supervision | VT/VT | VAT/VT | VAT/VAT |
|---|---|---|---|
| **GYT (Spatio-temporal)** | 16.2 | 16.8 | 17.0 |
| **YC-Inter (Spatial)** | 53.9 | 53.6 | 53.8 |

Table 11: **Effect of audio supervision in train and test**

**Number of frames for training.** We further evaluated the impact of different numbers of frames $T$ used for training. As shown in Table 10, selecting fewer frames for training significantly causes the performance to drop. We hypothesize that the model not only fail to capture the temporal dynamics with fewer frames but also loses some frames with groundable objects in the sentence while training. We also hypothesize that with a too large number of frames, more irrelevant frames might be selected during training, which decreases the performance.

**Effect of audio in training and testing.** Unlike text which describes a discrete concept as a target to ground, audio serves as a continuous representation that is highly relevant to the temporal information. For example, we can determine an action started when we hear a "cracking" sound. In Table 11, we tested our model using the additional audio modality. For the audio branch, we compute log-mel spectrograms and use a DAVEnet model (Harwath et al., 2018) initialized by MCN on HowTo100M (Chen et al., 2021) to extract audio features. We extend the global and local loss pairs from VT to VT, VA, and AT following (Shvetsova et al., 2022). We found when training and testing with audio, the spatio-temporal result increases the temporal performance while the spatial-only result remains the same. This validates our assumption that audio contributes more to temporal understanding. When we trained on audio and tested without audio, the performance increased over the VT model, showing that the audio serves as useful supervision for better video/text representations.

**Threshold for attention mask.** As shown in Figure 3(b), we apply a threshold to create a mask from the result of attention rollout. Note that this threshold $\tau$ is not a hyperparameter that affects the training or the model but simply serves as a means to an end to compute the *mAP* scores. We did not systematically optimize this threshold, but instead, Test different thresholds for attention scores for all relevant models (COMMA, ours) using the spatio-temporal grounding *mAP* IoU@0.4 on our GroundingYoutube dataset as shown in Table 12. We find 0.01 to be a reasonable threshold among all models, performing best on COMMA and giving at least the second best results for the proposed model.

## 10 GROUNDINGYOUTUBE ANNOTATION

We include more visualizations of our annotated GroundingYoutube dataset in video format at **sup/Annotation visualization videos.pptx**.

The data annotation was divided into three phases: During *Phase I* (Sec. 10.1, a graphical user interface (UI) and the task description were developed. In *Phase II*, the dataset was given to the annotators to generate the key points (Sec. 10.1). In *Phase III*, a manual quality control step was performed (Sec. 10.2).

### 10.1 DEVELOPMENT OF THE GRAPHICAL USER INTERFACE AND TASK DESCRIPTION

The annotation of a large amount of data is often one of the most expensive aspects of a machine learning pipeline design, which is why the annotation time per datum should be kept as short as possible. There are two points that can be optimized, (1) the training or the task "message" for the annotators and (2) the graphical user interface by minimizing interaction times.

| Treshold | Backbone | 0.1 | 0.05 | 0.01 | 0.005 | 0.001 |
|---|---|---|---|---|---|---|
| CoMMA* | S3D-word2vec | 0.76 | 0.90 | 0.93 | 0.91 | 0.86 |
| Ours | S3D-word2vec | 15.35 | 15.88 | 16.22 | 16.34 | 16.12 |
| CoMMA† | CLIP | 0.88 | 0.92 | 0.99 | 0.94 | 0.91 |
| Ours | CLIP | 15.93 | 16.33 | 17.10 | 17.05 | 16.24 |

Table 12: **Threshold for attention score on GroundingYoutube *mAP@0.4***

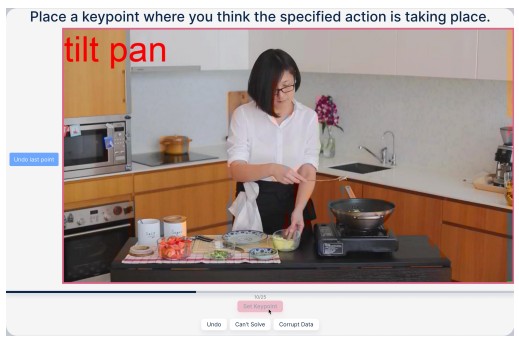

Figure 5: **A screenshot of our simplified annotation interface.** On the top, the annotation task is described in simple and short words to save reading time. To make interacting with the UI as intuitive as possible, actions are limited to simple button clicks and setting the key point by clicking on the image.

While tasks are usually formulated in such a way that no ambiguities arise, i.e. all possible edge cases are somehow covered, and simple words are used, in this case, we made a conscious decision to choose questions as short as possible, and that would give the annotator room for interpretation. We did this because it was hard to predict where people would actually locate actions in images. We also created a 1 min 30 sec long user training video where we demonstrate the task using exemplary keypoint annotations and explain how to use the UI.

Our annotation UI was designed with a special focus to keep it as intuitive as possible and reducing the interaction time. Our UI only provided five functionalities (set/unset a keypoint, undo the last image, image can't be solved, and image is corrupt) which were clearly described in text buttons (see Figure 5). Further, to reduce the cognitive load of our workers, images were presented in the form of work packages, each containing 25 images. Hence, we could ensure that completing a task would take no longer than 6 minutes.

The annotation of all $26,987$ images was performed with five distinct repeats per image, resulting in $134,935$ labels in total. All labels were generated by 13 professional annotators in total, which took them $5s$ in average per image. However, it should be noted that the number of images where an annotator placed a keypoint differs along all the workers (see Figure 6) and that the vast majority of all images have been answered by five annotators only. Examples are shown in Figure 7.

During the annotation, professional annotators were given a short instruction video at the beginning and then asked to click on the center of the given action without additional instructions. They were further free to choose "can't answer" if they could not locate the action, e.g., at the beginning and end of the clip. Thus, the number of available key points per image differs, and we choose majority voting to determine whether an action is present, resulting in new, refined temporal boundaries compared to the original annotation.

We found that the point-wise annotation resulted in roughly three distinct patterns, which depend on the captured scenario, as shown in Figure 8. In the case of half portrait or even wider shots in Figure 8a, annotations are highly locally centered. We further found that in some cases, the point annotation can also represent the flow of the action, e.g., pouring oil in Figure 8b, or even split into two separate clusters in Figure 8c.

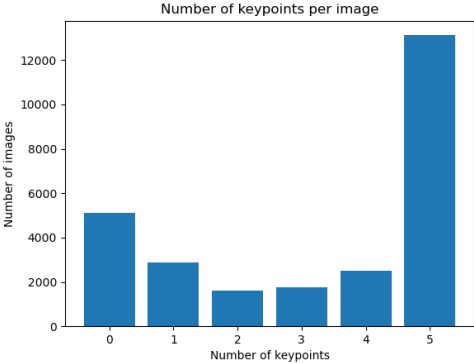

Figure 6: **Number of keypoints per image**. It can be seen that $48\%$ of the data has all 5 key points and $19\%$ has not a single annotation

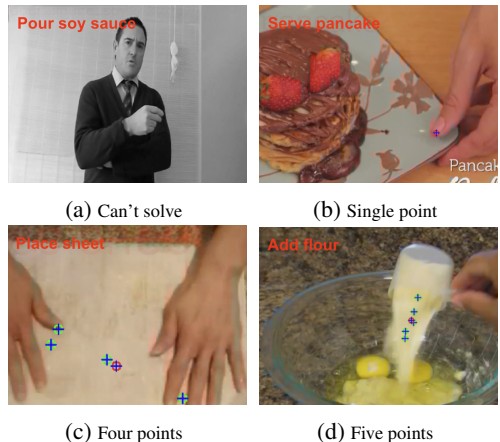

(a) Can't solve

(b) Single point

(c) Four points

(d) Five points

Figure 7: **Sample annotations**. The purple point represents the center point of the annotations in the frame. $48\%$ of the data has all 5 key points, and $19\%$ has not had a single annotation.

## 10.2 QUALITY CONTROL

Since the label quality of the datasets used is a critical factor in the performance of machine learning models, we verified the correctness of a subset of our images using an experienced annotation specialist for $1,026$ randomly selected frames. To evaluate the data quality, we evaluate the agreement between the annotation specialist and the annotations provided by the annotators. To this end, we considered an annotation as a false positive if three annotators or more have set a key point, although no action can be seen in the image, and as a false negative if three annotators or more have not set a key point, even though an action can be seen in the image. The entire sample was assessed using these criteria, with the specialist disagreeing with the annotators in only a total of $1.1\% \pm 3\%$ (FP: $0.7\% \pm 3\%$, FN: $0.4\% \pm 3\%$). We also found that annotations significantly diverted in terms of spread. Namely, wider shots tend to be highly centered, whereas zooming in together with the usage of larger objects such as a pan or a spatula results in more widespread key points. We also analyzed how often those cases occur and found that $14.0\%$ of the selected frames show a widespread pattern.

**Sample size calculation** To this end, we first needed a representative subset of $N_S$ images of our data. We calculated the required sample size based on the following two formulas:

$$N_0 = \frac{z^2}{\epsilon^2} \cdot p \cdot (1 - p) \tag{4}$$

where $\alpha$ is the confidence interval, $p$ the expected probability of the appearance of a quality aspect (e.g., widespread answers), $epsilon$ is the accepted error margin, and $Q(\alpha)$ is the percent point function of a normal distribution and $z = Q(1 - \frac{\alpha}{2})$.

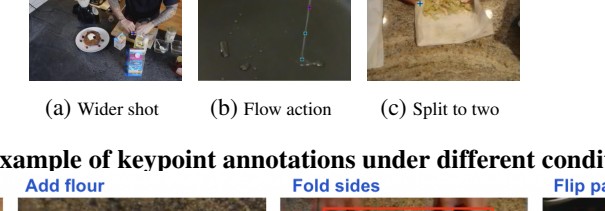

(a) Wider shot     (b) Flow action     (c) Split to two

Figure 8: **Example of keypoint annotations under different conditions.**

Saturated annotation     Flow annotation     Widespread annotation     Object interaction annotation

Figure 9: **Visualization on automatic bounding box generation from points.**

As $N_0$ would be the required sample size for an infinitely large population, we applied the finite population factor that results from sampling without replacement from a finite population.

$$N_S = \frac{(N_0 \cdot N)}{N_0 + (N - 1)} \tag{5}$$

where $N$ is the total number of images.

We set $\alpha = 95\%$, $epsilon = 3\%$, and our sample size of $N = 26{,}987$. As the probability of the quality aspect is unknown, we set $p = 50\%$, which resulted in $1{,}026$ being checked for quality control.

| Distribution Type | $mAP@0.4$ |
|---|---|
| Widespread actions | 18.34 |
| Saturated actions | 15.96 |
| Total | 17.10 |

Table 13: **Performance on the annotation distribution types of widespread v.s. saturated.**

### 10.3 DATASET USAGE FOR EVALUATION

**Bounding box generation:** For evaluation purposes, we get the union of all annotated points in a single frame with additional distance respect to the height $H$ and width $W$ as shown in Figure 9. We manually check the auto-generated bounding boxes and adjust the bounding box when needed.

**Performance on widespread and saturated action.** We evaluate the performance of different action distributions using the spatio-temporal grounding $mAP$ IoU@0.4 setting. We define widespread actions to have an area larger than a certain threshold $A$. Here, we set $A = 60{,}000$ pixels. As shown in Table 13, the performance of the widespread actions was higher since it had a higher tolerance of spatial localization error.

