# OpenReview forum: "What, when, and where? -- Self-Supervised Spatio-Temporal Grounding in Untrimmed Multi-Action Videos from Narrated Instructions"
_ICLR.cc/2024/Conference — ICLR 2024 Conference Withdrawn Submission_

### Official Review · Reviewer_MvS5 · 2023-10-25

**Soundness:** 2 fair
**Presentation:** 3 good
**Contribution:** 2 fair
**Rating:** 5
**Confidence:** 4

**Summary:**

This paper introduces an approach for learning spatio-temporal grounding with self-supervision and a new dataset. It proposes frame-wise alignment with local-global learning. Experiments show great results.

**Strengths:**

1. This paper is well-written and easy to follow.
2. The proposed dataset is interesting.

**Weaknesses:**

1. Novelty is limited. The proposed frame-wise matching is not new and has been well investigated in the temporal grounding field. Besides, the local-global learning is just borrowed from other similar tasks. The authors should carefully cite these previous works and provide discussion.

2. The experiments are not convincing. There is no statistics comparison between the proposed dataset and the existing datasets.

3. The compared methods are out-of-date. The authors should add more latest methods for comparison.

**Questions:**

1. Novelty is limited. The proposed frame-wise matching is not new and has been well investigated in the temporal grounding field. Besides, the local-global learning is just borrowed from other similar tasks. The authors should carefully cite these previous works and provide discussion.

2. The experiments are not convincing. There is no statistics comparison between the proposed dataset and the existing datasets.

3. The compared methods are out-of-date. The authors should add more latest methods for comparison.

---

> ### Author Response · Authors · 2023-11-13
>
> Thanks for taking the time to for reviewing the proposed work. We would have some question that would help us to better understand your points.
>
> #### Weaknesses/Questions:
>
> 1. *Novelty is limited:* Thanks for the suggestion. We have cited related works such as X-CLIP, Temporal Alignment Networks, ... etc. for related works. Could you please provide the related papers you have in mind for us to cite?
>
> 2. *Experiments are not convincing:* Please provide which part is not convincing. We will provide more details and justification. Also, we will create a table for statistical comparison.
>
> 3. *Compared methods are out-of-date:* We will add additional baseline GLIPv2 as suggested by reviewer tJVe29. Could you let us please know what methods you have in mind for comparison?

---

### Official Review · Reviewer_tJVe · 2023-10-30

**Soundness:** 3 good
**Presentation:** 3 good
**Contribution:** 3 good
**Rating:** 5
**Confidence:** 3

**Summary:**

The paper proposes a novel dataset called Grounding YouTube for spatial-temporal grounding in untrimmed videos. It further proposes a self-supervised learning approach for spatio-temporal action grounding trained on loose video and subtitle supervision only, outperforming a few self-supervised baseline approaches.

**Strengths:**

1. The paper proposes a novel dataset that combines both spatial and temporal grounding. Annotation details and evaluation protocols are clearly presented. Quality control indicates that the dataset is likely to have a high quality.

2. The paper compares a few self-supervised baseline methods on the proposed dataset and show the proposed method outperforms all of them.

**Weaknesses:**

1. ActivityNet-Entities [1] is an extension of ActivityNet-Captions that contains grounding annotations. This dataset seems quite relevant to what this submission is tackling. The submission might need to discuss ActivityNet-Entities and provide experimental comparison on this dataset if necessary.

2. "Pointing game accuracy" is not a commonly known metric. To make the submission more self-contained, a citation might be needed - firstly proposed in [2] if I recall correctly.

3. The description for "Mining: MLP" (1st row in Table 4) is missing, though it is referring to the same paper with MIL-NCE.

4. How is IoU+Pointing game computed? Is it simply an addition between IoU and the Pointing game accuracy? The authors may need to provide more details about how the metric is combined and the rationale.

5. Recent progresses on spatial grounding and temporal grounding are missing, to name a few: [3,4,5]. Particularly, my hunch is that the recent advances of open-vocabulary object detector may push the limits of spatial grounding by quite a lot since the objects in videos may appear often existing large-scale image grounding/detection datasets.

[1] Zhou, et al. "Grounded video description." CVPR 2019

[2] Zhang, et al. "Top-Down Neural Attention by Excitation Backprop". ECCV 2016

[3] Zhang, et al. "Glipv2: Unifying localization and vision-language understanding." NeurIPS 2022

[4] Yao, et al. "Detclip: Dictionary-enriched visual-concept paralleled pre-training for open-world detection." NeurIPS 2022.

[5] Chen, et al. "End-to-end Multi-modal Video Temporal Grounding". NeurIPS 2021

**Questions:**

My questions are already listed in the weakness section (see above).

---

> ### Author Response · Authors · 2023-11-13
>
> Thanks for taking the time to for reviewing the proposed work and for providing valuable feedback! We try to address the raised points in the following:
>
> #### Weaknesses:
>
> 1. *Comparison with ActivityNet-Entities :*  ActivityNet entities actually focuses only on objects in the video, providing bounding boxes exclusively for noun phrases, while our training and evaluation are action-focused, such as 'cutting apples' and 'frying eggs'. We propose to a respective discussion in the related work, but we think that ActivityNet-Entities might not be a good benchmark for the spatial-temporal grounding of actions. (Please see also our answer to Reviewer CNqU)
>
> 2. *Pointing game metric:* Thanks for the comment. The pointing game is a relevant metric for grounding in the detector-free setting see e.g. [1, 2] and many more, where the bounding box proposal wasn't used/provided. We will add the citation to the paper.
>
> [1] Arbelle, Assaf, et al. Detector-free weakly supervised grounding by separation. ICCV. 2021.
> [2] Eyal Gomel, et al. Box-based Refinement for Weakly Supervised and Unsupervised Localization Tasks, ICCV, 2023
>
> 3. *Description missing:* Thanks for the catch. We will update the reference.
>
> 4. *How is IoU+Pointing game computed? :* We include the details in the supplement section 8.3 Evaluation metrics. IoU+Pointing game combines the evaluation setting from the spatial grounding (Akbari
> et al., 2019) and temporal grounding (Kuehne et al., 2019) metrics. For each video frame, the prediction is correct when the model predicts the correct action for the frame. Also, given the predicted action as a query, the maximum point of the heatmap aims to be within the desired bounding box.
> We then computed the Intersection over Union (IoU) over all the predictions with the GT to acquire
> the final score.
>
> 5. *Missing references/baselines:* Thanks for the suggestion! We found that [5] doesn't provide public code, and [4] achieved similar performance as one of our baseline GLIP. We will include GLIPv2 as our additional baseline.

---

### Official Review · Reviewer_YqMJ · 2023-10-31

**Soundness:** 3 good
**Presentation:** 3 good
**Contribution:** 2 fair
**Rating:** 5
**Confidence:** 5

**Summary:**

The paper addresses the task of spatio-temporal grounding, which involves localizing events in space and time within video data based on verbal descriptions. The authors propose a new approach that uses multimodal supervision, specifically video and subtitle data, without the need for human annotation. Specifically, they introduce a combination of local representation learning (for fine-grained spatial information) and global representation encoding (for higher-level representations). They also present a new benchmark dataset for evaluating spatio-temporal grounding in long, untrimmed, multi-action instructional videos. The proposed method outperforms current baselines in various settings.

**Strengths:**

1. The paper introduces a novel method for spatio-temporal grounding using multimodal supervision. This approach, which leverages both video and subtitle data without the need for human annotations, is a step closer to real-world senarios than the traditional methods that rely heavily on human-annotated sentences and bounding box supervision.
2. It is interesting to apply Sinkhorn-Knopp Optimal transport algorithm for key frame selection. If would be good if the authors could give some visualization results to show the effectiveness of this strategy. For example, showing the key frames selected by the algorithm.
3. The authors have thoroughly evaluated their proposed method against other state-of-the-art techniques on both the new dataset and standard tasks. Their method's ability to outperform current baselines in various settings adds credibility to their approach.

**Weaknesses:**

1. The paper's approach to the global feature seems to primarily focus on aggregating frame-level appearance features. However, it does not appear to adequately model the temporal relation between key frames. This might pose a challenge in distinguishing actions driven by hand motions, such as 'pick up' and 'put down'. Could you provide insights into this aspect?
2. The choice to use key points instead of bounding boxes for the proposed dataset raises concerns. Key points can be inherently subjective, leading to inconsistencies in annotations. This subjectivity might affect the reliability and generalizability of any conclusions drawn from this dataset.
3. The rationale provided for certain choices, such as the preference for key points over bounding boxes due to distractions by object outlines, is not entirely convincing. A more robust justification or empirical evidence supporting such decisions would strengthen the paper's arguments.

**Questions:**

1. Could you provide visualization results to demonstrate the effectiveness of the Sinkhorn-Knopp Optimal transport algorithm in key frame selection? It would be particularly insightful to see the key frames selected by the algorithm.

2. The paper's approach to the global feature seems to focus on aggregating frame-level appearance features without adequately modeling the temporal relation between key frames. How does the model address the challenge of distinguishing actions driven by hand motions, such as 'pick up' and 'put down'?

3. Could you provide a more detailed justification or empirical evidence for the preference of key points over bounding boxes, especially in light of potential distractions by object outlines?

---

> ### Author Response · Authors · 2023-11-13
>
> Thanks for highlighting the strengths of the proposed paper and for providing valuable feedback! As all weaknesses are complemented with respective questions, we will focus on those specifically.
>
>
> #### Questions:
> 1. *Visualization to demonstrate the OT for key frame selection?:* Thanks for the suggestion! We will work on the visualization and report it soon.
>
> 2. *Modeling the temporal relation between key frames:* Thanks for the question. We would like to answer in multiple ways here: First, we actually evaluate the framework with both, 3D and 2D backbones in Table 2, namely a S3D backbone which captures low-level temporal dynamics as the requested ones. Table 2 also shows that even while CLIP does not capture such temporal dependency it still performs slightly better than the S3D backbone. This can be considered a phenomenon known from various current video approaches using CLIP backbone, such as [1,2]. To address this issue, we offer to include an additional experiment with the TimeSformer backbone, which also utilize a visual transformer that encodes temporal dynamic, for the camera-ready version.
> To reason why CLIP might perform better than the S3D backbone, we want to point out that our dataset is targeted to be used for grounding instead of classification. Hence, our text queries seldom contain explicit complementary semantics such as pick up and put down as it would be labeled in action classification tasks.
>
> [1] Luo, Huaishao, et al. "Clip4clip: An empirical study of clip for end-to-end video clip retrieval and captioning." Neurocomputing 508 (2022): 293-304.
> [2] Bain, Max, et al. "A CLIP-Hitchhiker's Guide to Long Video Retrieval." arXiv preprint arXiv:2205.08508 (2022).
>
> 3. *Justification or empirical evidence for the preference of key points:* We have conducted an annotation study in supplement Figure 6,7,9. Our finding is that our annotation reaches high consistency and captures meaningful variety for evaluation. In general we found in the initial phase of the annotation that the center point of action is more intuitive than the bounding box since it is hard to give any guidance of the scale of the bounding box, thus the physical outline for an action, which varies e.g. with respect to annotations and viewpoints of the hands, human, or instrument. As an alternative view, if we would predefined certain label criteria (e.g. always include the hands), the model would not be aware of those criteria without being trained on it, which would defy the idea of a weakly-supervised training without the need for annotation. Instead, we used the current spread of center points to define the outlines of the region that we want the model to localize on.
>
>
> Please let us know if you have any other questions or concerns regarding our paper or response.

---

### Official Review · Reviewer_CNqU · 2023-11-01

**Soundness:** 2 fair
**Presentation:** 3 good
**Contribution:** 2 fair
**Rating:** 3
**Confidence:** 5

**Summary:**

This work tackles the challenging problem of spatio-temporal video grounding (STVG). They propose a framework trained on loose "annotations", where the ASRs, extracted from the videos, are used to align the multiple modalities. Although transcripts could provide a "free" source of information for STVG, they come with a significant challenge. ASR is noisy for various reasons, and not all the comments in the video are necessarily related to what is happening at that moment or even in the video itself. In this framework, they propose to combine global and local representations to learn STVG. They also present a frame selection strategy with optimal transport to deal with the loose annotations, in addition to a new benchmark dataset, which provides dense annotations in long, untrimmed, multi-action instructional videos.

**Strengths:**

- Challenging problem that is very important for the video analysis community.
- Although the Global and local representations have been explore in other works, the method is reasonable.
- In general the paper is well written.

**Weaknesses:**

- ActivityNet has a variant called ActivityNet entities (Grounded Video Description, Zhou, L. et. al.), which contains temporal and spatial annotations. It would be great to see the performance of this method on that dataset since it is a well-study benchmark for temporal grounding.

-  Since ActivityNet-Entities exists, I found the claim in section 4 "current downstream datasets either provide spatial, temporal annotations or short video clip with spatio-temporal annotations. These datasets do not provide the opportunity to evaluate both aspects, spatial and temporal grounding, in an untrimmed long video manner." not true.

- I found the notations confusing. Please see question 4.

**Questions:**

1. Why can we not use ActivityNet as a benchmark for STVG?

2. How long are the videos in the GoundingYoutube benchmark? Since I don't know how long the videos are, it is difficult to suggest how precise the method is. From Table 2, we can infer that the method has good recall and low precision. But what is the nature of the dataset? It could be that all the temporal groundings are happening in a certain portion of the video or by just predicting the beginning and endinding of the video as the temporal location is good enough for a 0.1 threshold (we have seen this in other datasets). Please look at Figures 2 and 4 of  "A Closer Look at Temporal Sentence Grounding in Videos: Dataset and Metric"

3. In the temporal grounding literature it is well established the tIoU@thresholds metrics and the mean_tIoU to evaluate the predictions, other spatial-temporal grounding works like HC-STVG use tIoU @ thresholds to evaluate the performance of the predictions. Can you evaluate the performance of your method using that metric? and take a look a  "A Closer Look at Temporal Sentence Grounding in Videos: Dataset and Metric" which weighs the low IoU thresholds. I would like to see the bands 0.1, 0.3, 0.5, 0.7 and 0.9.

4. Section 3.1 "t \in {1,...,T} represents the number of frames in the video". Section 3.2, "our goal is to find T frames out of the U frames". can you please clarify this? is U or T the total amount of frames in the video? are those frames sampled from the video at what fps? 5?

5. About the frame sampling strategy, I recommend looking at "DORi: Discovering Object Relationships for Moment Localization of a Natural
Language Query in a Video" where they use the sharpest frame in a small clip. It could help to get better frames for Sinkhorn-knopp alignment. I also think is missed related work since they don't use Bbox annotations.

---

> ### Author Response · Authors · 2023-11-13
>
> Thank you for reviewing our paper, pointing out the strengths of our work, and providing valuable feedback! We hope to address some concerns in the following:
>
> #### Questions:
> 1. *Performance on ActivityNet entities - Why can we not use ActivityNet as a benchmark for STVG? :* ActivityNet entities actually provides bonding boxes for noun phrases in the captions, namely the objects. It is thus closer related to an object detection task and does not capture any spatial or temporal extent of actions. As our setup and evaluation are action-focused, such as cutting apples and frying eggs, ActivityNet entities might not be a good benchmark here. We offer to include ActivityNet entities and to add the comparison to clarify this. (Please see also our answer to Reviewer tJVe)
>
> 2. *How long are the videos in the GoundingYoutube benchmark? :* Our annotation is based on the MiningYoutube dataset, where each video is around 3-5 minutes with an average of 41.6 labeled action instances per video. Note that each video contains various classes. We will clarify this in the revision.
>
> 3. *Evaluate the performance of your method using m_tIOU and tIOU@?:* Thanks for the comment. The mIOU metric reported in Table 2 is the same as the vIoU in the HC-STVG paper. We will clarify this in the paper. The HC-STVG paper did not report tIoU@ with various thresholds, but we agree that this could be a valuable add-on. We currently work on the tIoU numbers and hope to report them soon.
>
> 4. *Notation of T in Sec 3.1 : =* Thanks for the catch. The T in Section 3.1 should be U. We will correct this typo. The frames were sampled with 5 FPS mentioned in the supplement section 9.1. We will clarify this in the updated version.
>
> 5. *Sharpest frame sampling:* Thanks for the suggestion. It is worth mentioning that we have evaluated the number of frames to sample, including the single-frame strategy, in supplement Table 10. We find using a single frame for training results in low performance due to the lack of temporal dynamics. We further assume that technically, the frames with significant blur should have low visual language similarity, and will therefore not be selected during our training. To validate this intuition, we propose to evaluate the sharpest frame strategy for the camera-ready version as it requires training of the system.